**Perspective**

# Understanding long-term physical and psychosocial outcomes from conflict to rehabilitation through the ADVANCE cohort

Fraje Watson[1,11], Oliver O'Sullivan[2,3,11] ✉, Rabeea Maqsood[1], Daniel Dyball[4], Fearghal P. Behan[5], Anna Verey[4], Howard Burdett[4], Sarah Dixon Smith[6], Gareth Evans[7], Warren Allison[8], Susie Schofield[9], Emma Coady[1], Eleanor F. Miller[1], Anthony M. J. Bull[1], Paul Cullinan[9], Nicola T. Fear[4], Christopher Boos[1,10], Harriet Kemp[6] & Alex N. Bennett[1,2]

Throughout the history of conflict, medical advancements have improved people's survival. Many individuals sustain physical and mental scars, often unseen and impacting on future health. Improvements in clinical care are often translated into practice, but long-term health impacts may not be appreciated until later. Most post-conflict data related to health outcomes of military personnel are limited by retrospective methodologies examining isolated results or surrogate measures of health. The ADVANCE cohort aims to prospectively understand the long-term physical and psychosocial outcomes of conflict-injured UK Service personnel who served in Afghanistan compared to an uninjured comparison group. This review will outline cardiovascular, mental health and musculoskeletal findings in this cohort, discuss the evolution of the objectives to understand associated mechanisms and mediators, and demonstrate the initial impact and relevance for military and civilian populations worldwide. Finally, two participants reflect on their involvement in the ADVANCE cohort and its impact from their perspective.

The history of conflict medicine follows incremental changes in life-saving interventions, surgical practice and rehabilitation medicine[1] with some contribution to survival being credited to improvements in personal protective equipment[1,2]. Following improvements in initial survival of conflict, there is a considerable burden of mental health and musculoskeletal sequalae[3,4]. Additionally, veterans are more likely to have future cardiovascular morbidity and mortality[5]. This phenomenon has been seen in retrospective studies examining combat survivors of the First World War[6,7], the Vietnam War[8,9], the 1991 Gulf War[10], and in more recent conflicts[11]. The conflict in Afghanistan resulted in many 'unexpected survivors'. This term has been used to describe individuals who sustained severe injuries, such as extremity injuries, that, in previous engagements, would have resulted in

their death[12]. Whilst their short-term outcomes were good, the long-term health consequences of such severely injured personnel remain largely unknown[13–17].

Assessing long-term health outcome data in Service personnel following conflict can be challenging as data is mostly retrospective, focused on a single outcome, underpowered, uses surrogate outcomes such as return to active duty, or is not specifically related to conflict trauma[18]. It is also important to note that negative physical and psychosocial outcomes can be associated with military Service, regardless of deployment[19]. Therefore, it is appropriate to use a military comparison group, that did not sustain such an injury, to truly assess the long-term outcomes of conflict injury. Furthermore, the trauma, rehabilitation, and social care received in the 21st century

[1]Bioengineering, Faculty of Engineering, Imperial College London, London, UK. [2]Academic Department of Military Rehabilitation, Defence Military Rehabilitation Centre, Stanford Hall, Stanford, UK. [3]Academic Unit of Injury, Recovery and Inflammation Sciences, School of Medicine, University of Nottingham, Nottingham, UK. [4]King's Centre for Military Research, King's College London, London, UK. [5]Discipline of Physiotherapy, School of Medicine, Trinity College Dublin, Dublin, Ireland. [6]Department of Surgery and Cancer, Faculty of Medicine, Imperial College London, London, UK. [7]Participant partner, Surrey, UK. [8]Participant partner, Wiltshire, UK. [9]National Heart & Lung Institute, Faculty of Medicine, Imperial College London, London, UK. [10]Faculty of Health and Social Sciences, Bournemouth University, Bournemouth, UK. [11]These authors contributed equally: Fraje Watson, Oliver O'Sullivan. ✉e-mail: oliver_osullivan@hotmail.com

has progressed, and outcomes from previous conflicts are no longer representative of today[20,21]. Despite overall advancement in medical care, global conflicts continue to result in death and severe physical injury of both military personnel[20] and civilians[21], making the understanding of long-term health consequences an international priority[22,23].

A prospective longitudinal cohort detailing the long-term outcomes of conflict injury was therefore warranted. The ArmeD serVices trAuma rehabilitatioN outComE (ADVANCE) cohort was established in 2013 to compare long-term physical and psychosocial outcomes of UK Service personnel and veterans who served in the Afghanistan conflict on combat operations between 2003 and 2014. Over 70% of ADVANCE injuries were due to blast, the most common mechanism of injury sustained by civilians in contemporary conflicts[24]. Extremity injury is also well represented in ADVANCE, and this type of injury continues to be particularly prevalent in current conflicts, such as the Russo-Ukrainian war[25,26].

This article aims to provide an overview of novel findings, clinical practice and research implications from the ADVANCE cohort and the future direction of exploration within this unique group. It will explain the initial ambitions of ADVANCE, briefly describe outcomes of the core objectives, and consider the responsibilities of a long-term observational study. Guided by current findings, the core objectives have evolved to explain the mechanisms and mediators influencing the reported outcomes using novel technology and techniques. This evolution will be described alongside a discussion on the dataset's interconnectedness and impact on policy and practice. Finally, representative participants will discuss their opinions, experiences and reflections on the ADVANCE cohort.

## Who are the ADVANCE cohort
Between 2016 and 2020, the ADVANCE cohort recruited 1145 male participants who served in the UK military in the Afghanistan conflict; approximately half of whom sustained serious physical conflict injuries (Injured; $n = 579$), and a comparison group, frequency matched for age, deployment period, Service, rank, regiment, and role-in-theatre (Uninjured; $n = 566$) (Fig. 1). Potential participants were identified from Service records (including deployment records and medical histories) and contacted via multiple methods[18].

Ethical approval was granted in 2013 by the UK Ministry of Defence Research Ethics Committee (MODREC: 357PPE12), and recruitment was started in March 2016 with the Baseline (median 8-years post-injury/deployment) and Follow-up 1 (median 11-years post-injury/deployment) data collections completed in August 2020 and 2023, respectively, with Follow-up 2 (minimum 6 years post baseline and at least 2 years post follow-up 1) underway. Participants attend a full study-day at the Defence Medical Rehabilitation Centre, which includes a Nurse-led clinical interview, medical assessment and self-report questionnaires (Fig. 1)[18].

At the point of injury/matched deployment, participants were a median of 25.5 years old, and at Baseline, participants were a median of 34.1 years old. Full cohort demographics can be found here[27]. Limb loss is considered the signature injury of the Afghanistan conflict[28], which was sustained by 161 ADVANCE participants. Therefore, analyses were often conducted for the whole Injured cohort as well as studying those with limb loss specifically.

## Cohort retention
Participant retention is essential to the success and validity of longitudinal cohorts[29]. The ADVANCE cohort utilises multiple retention strategies, including compensation, ease of access and communication/tracing strategies (Fig. 2), resulting in a retention rate of 92% between Baseline and Follow-up 1. Attendance at Follow-up 2 is currently at 61% and due for completion in summer 2026. Broader engagement with the cohort is maintained through regular briefings with veterans' charities and services such as Help for Heroes, the Royal British Legion, BLESMA, the CASEVAC Club and OpRESTORE.

## Core objective outcomes
The core objective of the ADVANCE cohort is to investigate the long-term physical and psychosocial outcomes of UK Service personnel and veterans who sustained serious physical conflict injuries during their deployment to Afghanistan, compared to a frequency-matched comparison group[18]; Injured and Uninjured groups as previously described. This ambition encompassed the core outcomes of cardiovascular disease and risk, musculoskeletal and mental health.

The main discoveries from the core objectives across each of these themes thus far are described below and illustrated in Fig. 3. Since the core objectives were decided, the ADVANCE study has evolved to collect additional data on new and existing themes, conducted cross-theme analyses, and new directional hypotheses to expand on findings from the core analysis. Findings from the core objectives will be described here, and results from further work will be discussed later.

### Cardiovascular
Measurements obtained during the physical exam included arterial pulse wave form analysis, heart rate variability, venous blood samples, and spirometry[18]. At Baseline, a higher resting heart rate and lower heart rate variability was identified in the Injured group compared to the Uninjured group, indicative of autonomic imbalance[30,31]. Metabolic syndrome increased arterial stiffness and reduced resting estimated myocardial blood flow, which was more prevalent in the Injured compared to the Uninjured group[32,33]. These differences were even more pronounced among those with more severe injuries, including limb loss[33,34]. These findings may be explained by increased relative abdominal obesity and visceral fat, low-grade systemic inflammation and lower physical function in the injured versus uninjured groups, or additional factors[30–32].

### Musculoskeletal
To evaluate musculoskeletal health, the physical examination included knee and hip radiography, full-body dual-emission X-ray absorptiometry, and self-reported questionnaires[18]. Musculoskeletal findings demonstrated worse hip and knee osteoarthritis[27,35], upper limb disability[36] and increased reported low back pain[37] in the injured compared to the uninjured group. Those injured with lower limb loss had worse knee osteoarthritis, upper limb disability, bone mineral density, and low back pain than the Uninjured group[27,36–38]. Between visits, an increased incidence and progressive knee osteoarthritis was observed for individuals with lower limb loss compared to the uninjured group. We hypothesise that this represents a change in the primary cause of osteoarthritis development in this subgroup from post-traumatic (related to the acute injury and immediate aftermath) to bio-mechanical/mechanoinflammatory (related to long-term walking changes), with implications for clinical care[39,40].

### Mental health
Using self-reported questionnaires, we found an increased likelihood of generalised anxiety, depression, post-traumatic stress disorder, suicidal ideation and mental health multimorbidity in the Injured group compared to the uninjured group[41,42], even though both groups are likely to have witnessed conflict events (e.g. death or injury of friend/colleague, small arms fire). Qualitative research discovered that mental health needs were often not actively addressed during physical recovery, treatment pathways could be inflexible, and continuity of care following medical discharge and transfer to civilian healthcare providers was sometimes poor[43,44].

## Outcomes for participants with limb loss
Throughout the core objective analysis, a theme emerged amongst Injured group individuals with limb loss; they had worse physical outcomes but better mental health than those Injured without limb loss. For example, they had more severe knee and hip osteoarthritis than the Uninjured group but reported no differences for knee and hip outcome scores assessing factors such as function, pain, disability[27,40,45]. Those injured without limb loss had

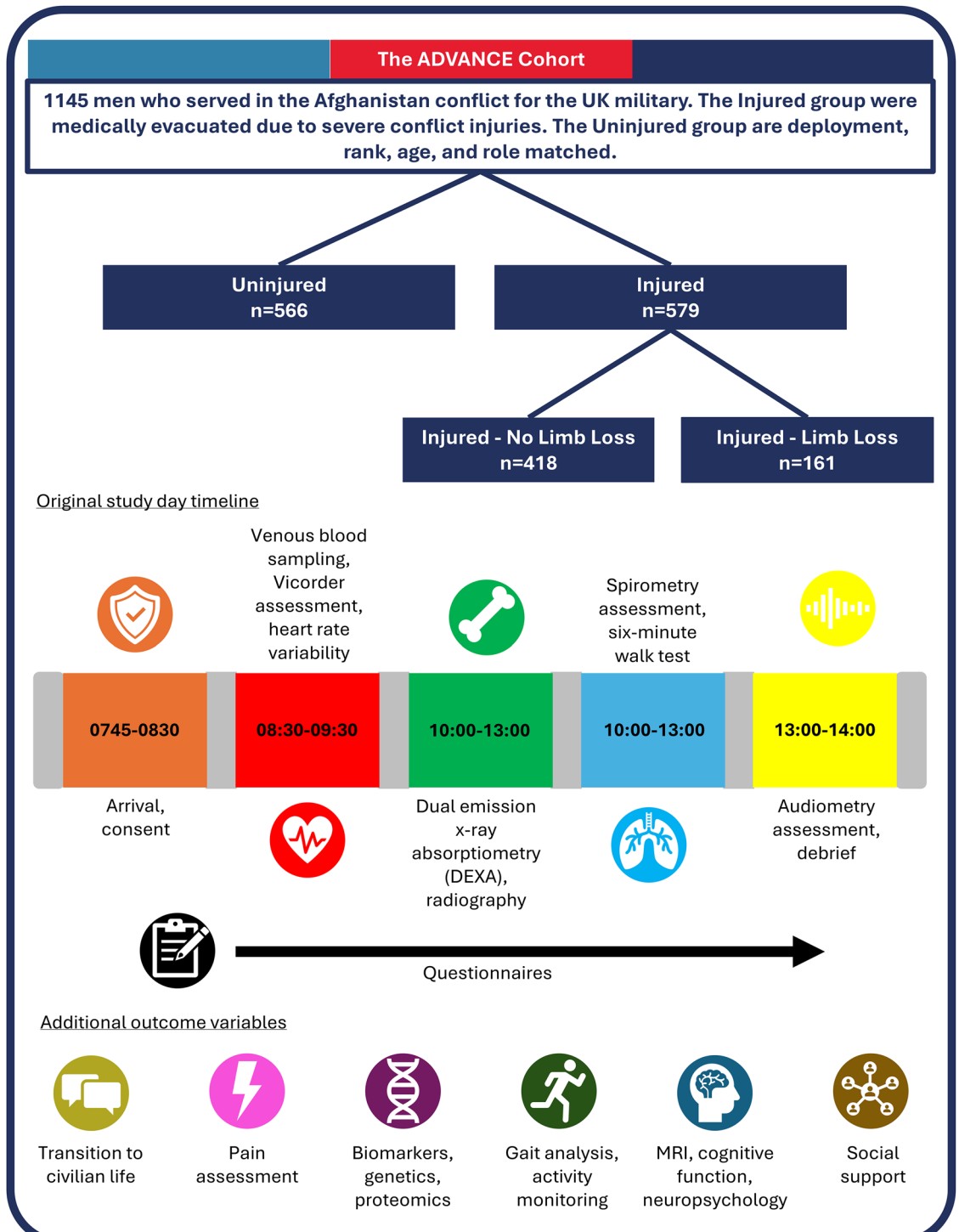

**Fig. 1 | The ADVANCE cohort.** The cohort, sub-groups, data collection schedule and example outcome variables included in the ADVANCE study.

worse mental health outcomes compared to the Uninjured group, whereas those injured with limb loss had similar mental health as the uninjured group[41,42]. Furthermore, those with limb loss reported a large degree of post-traumatic growth, whereby serious trauma is followed by beneficial psychological effects[46].

A 'hierarchy of wounding' has been hypothesised, which could lead to more positive outcomes for those injured with limb loss and negative outcomes for those injured without limb loss[47]. Some of the postulated drivers for this include the visibility of limb loss injuries, cultural context in the UK (severely injured veterans had increased visibility and accessibility to

opportunities), and financial security as a result of limb loss-related compensation[44,47]. In addition, intensive medical care led to improvements in perceived health, with continued and regular interaction with healthcare professionals for limb care services, or more physical and psychosocial support during their initial rehabilitation, which they considered timely and effective[43]. Social support, as measured by the Multidimensional Scale of Perceived Social Support, was similar in the Injured and Uninjured groups. We did not observe a difference in those with and without limb loss, but moderate-high levels of social support were associated with better mental health outcomes overall[48].

- Access to healthcare data in the form of a 'health MOT'
- Ease of access:
  - provision of a travel allowance
  - accommodation provision
  - communication with employers to allow individuals to take medical leave to attend appointments
- Communication:
  - regular communications with participants through newsletters, social media posts, birthday cards and Christmas cards
- Tracing:
  - participants are reminded and encouraged to update their contact information
  - 'best contact' details provided by the participant in case their details lapse
  - use of electoral roll data to re-establish contact with the participant
- Financial compensation

**Fig. 2 | Retention strategies utilised in the ADVANCE cohort.** The ADVANCE cohort has a 92% retention rate between Baseline and first Follow-up data collections using the strategies listed below.

## Impact of ADVANCE so far

Traditional academic promulgation has led to this work being shared at national and international forums, e.g.[45,49] and discussed in Editorials, e.g.[47,50]. These findings have influenced care pathways and resource allocation at multi-departmental UK Governmental level, including lessons from ADVANCE being applied via the Defence Medical Rehabilitation Centre to medical staff treating casualties from the Ukrainian conflict (https://www.gov.uk/government/news/uk-steps-up-life-saving-medical-support-for-ukraines-armed-forces). Currently, these findings are leading to collaborations with national third-sector organisations and international research groups, e.g. the World COACH Consortium[45].

Within the UK military, ADVANCE findings have led to direct improvements in clinical care through the Military Osteoarthritis Group, which was created to develop specific prevention and management strategies for Service personnel with osteoarthritis, supported by medical policy[51]. Additionally, in accordance with ethical research principles and with the participant's consent, when abnormal research findings are detected, these are highlighted to their doctor[52]. This now includes any electrocardiogram abnormalities and prospective risk of stroke or heart attack within 10 years.

Clinical interventions are also being piloted within the ADVANCE cohort, led by earlier findings demonstrating localised bone mineral density loss in participants with lower limb loss due to reduced biomechanical loading[38]. Exercise interventions utilising specific loading protocols have improved bone mineral density in people following space flight, post-menopause, and in those recovering from anorexia[53]. However, existing protocols are inappropriate for people with lower limb loss due to joint loss, prosthetic interface, reduced balance, and loading via socket prosthesis. A Delphi process investigated what an appropriate exercise intervention to minimise bone mineral density loss for people with lower limb loss could look like[53,54]. These include initially supervised interventions performed at least twice a week for 6 months, including weight-bearing, multi-planar and high-impact exercises[54]. These efforts have led to the design of a feasibility trial that will be conducted within a small sub-cohort of ADVANCE participants with lower limb loss before a full intervention study is developed in a general population of those with lower limb loss.

## Evolution of the ADVANCE cohort

Over the course of ADVANCE, the core outcomes of cardiovascular, musculoskeletal and mental health have been supplemented as the cohort has developed. Recruitment started nearly a decade ago, and the cohort is approaching completion of its third data collection timepoint. In this time, interest in this unique cohort has led to increased collaboration to investigate novel research areas (e.g., MRI brain scans for traumatic brain injury, biomechanical gait assessment, DNA methylation analysis for genetic ageing assessment, field-hospital practice and patient-reported outcomes for pain assessment[39,55]). Furthermore, accessibility and availability of technological advancement, laboratory analysis costs, and computing power, has allowed for additional outcomes to be measured (e.g. wrist-worn activity monitors, advanced molecular analysis, and machine learning techniques[56]) (Fig. 3). In recent years, an Independent Scientific Advisory Group (https://www.advancestudydmrc.org.uk/isag/) has been recruited to support, advise and challenge the processes within ADVANCE.

## Additional data sources

In addition to new themes, ADVANCE has worked to align multiple data sources from across the Ministry of Defence to enable a greater understanding of the entire patient journey and how this might influence future outcomes (Fig. 4). These sources provide pre- and field-hospital intervention data, enhance existing Joint Theatre Trauma Registry injury data, and record post-injury physical and psychosocial rehabilitation.

Additional data linkages enhance understanding of the impact of immediate and early medical and surgical interventions on long-term physical and mental health, with implications for clinical policy and practice. For example, analysis of the Ministry of Defence's Joint Theatre Trauma Registry and Medical Emergency Response Team datasets (Fig. 4) revealed prehospital rapid anaesthesia is associated with lower levels of mental health need, suggesting the procedure may have a protective role against psychological distress[57]. In contrast to current clinical tenets, investigation of the Deployed Blood Transfusion Databases and (super) massive blood transfusions found no evidence of increased inflammation or other negative impacts on cardiovascular health in recipients; in fact, participants who had received massive transfusions demonstrated lower levels of cholesterol, which may be related to improved endothelial function post-transfusion[58]. The detailed injury data provided by the logbooks and Joint Theatre Trauma Registry has allowed an analysis of polytraumatic blast injury pattern 'constellations' in 400 of the Injured participants, thereby helping to improve traumatic injury scoring systems and provision of personal protective equipment[59].

These data were being used in current projects to explore the injury, anaesthetic, and psychosocial factors associated with the development of persistent pain, to improve triage tools to identify those most at risk and determine if specific point of injury anaesthetic or analgesic interventions are associated with improved outcomes.

## Collaboration and future research in ADVANCE to ask "why?", "how?", and "so what?"

As a result of the ADVANCE cohort evolution, we have developed deterministic hypotheses, collaborated across themes, and generated ideas for future research and intervention development. In doing so, we aim to harness the power of a multi- and inter-disciplinary network of researchers to understand the mechanisms and mediators of the findings, thus improving the ability to identify, explain, and manage the long-term health sequelae of conflict-related injury (Fig. 3). Here we provide examples of this work.

From a cardiovascular perspective, our preliminary research suggests that physical function may have a positive mediatory effect and may offset the impact of conflict injury on heart rate variability[60]. We have also identified a relationship between arterial stiffness and heart rate variability, independent of injury and traditional cardiovascular risk factors[61]. Slow-paced breathing appeared to improve heart rate variability levels more than normal breathing in the injured group[62,63]. Together, these findings support

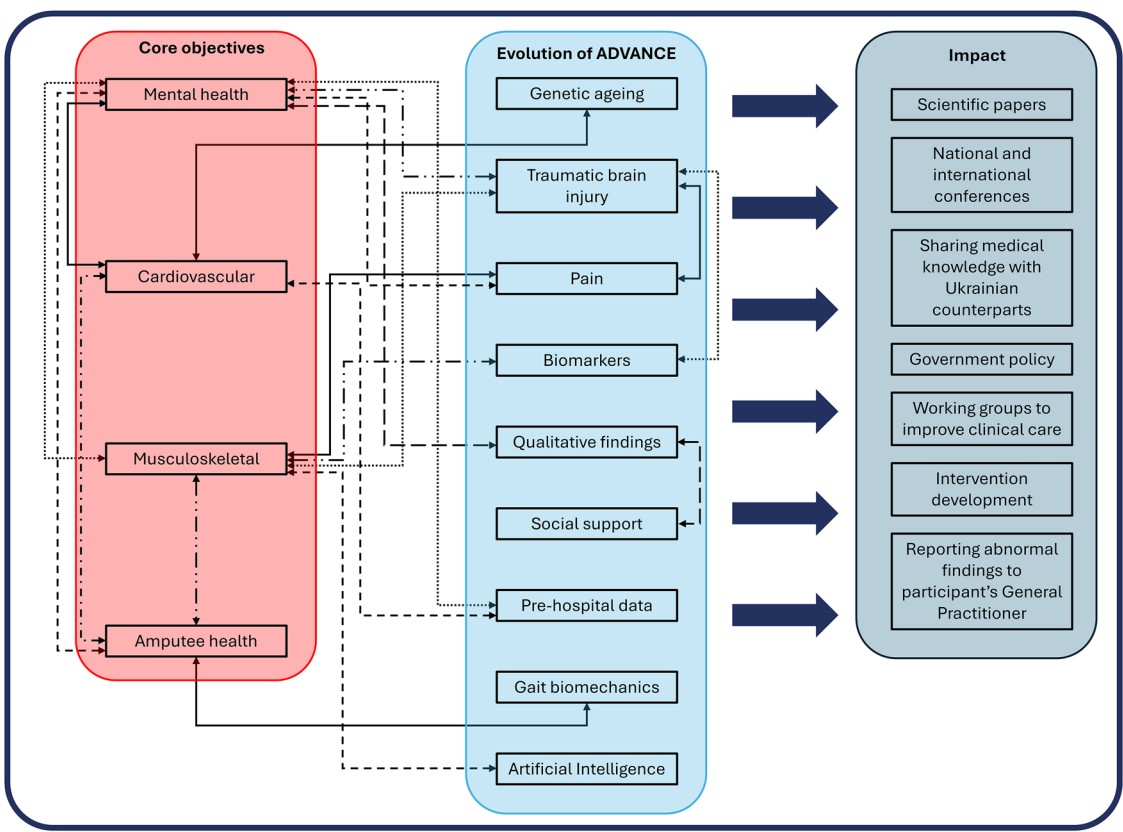

**Fig. 3 | The evolution of the ADVANCE study objectives and impact.** Infographic detailing how the core objectives, new themes and collaborative efforts lead to impact.

**Fig. 4 | Additional data sources are available to the ADVANCE cohort.** These data sources included pre-hospital and hospital sources providing valuable information on the acute injury and treatment.

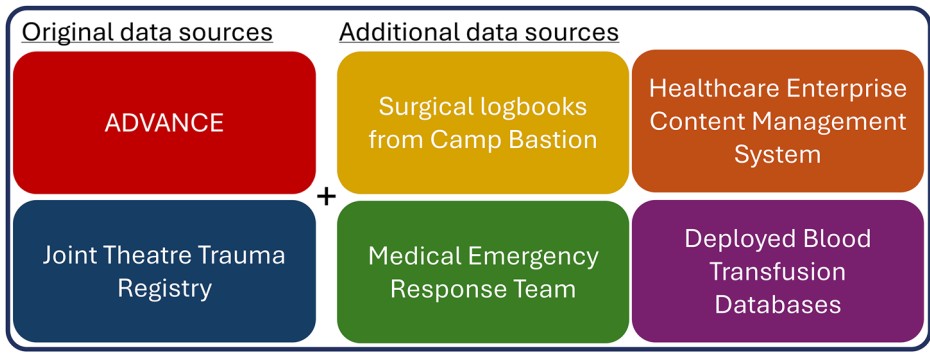

the design of self-administrable and non-invasive heart rate variability-biofeedback training to reduce future cardiovascular risk[61]. This will be piloted in a subset of the ADVANCE cohort to assess its feasibility and effectiveness. If successful, it could be cost-effectively applied to other populations affected by conflict injury or similar trauma and may lead to improvement in cardiovascular health, autonomic function, stress, and overall quality of life.

The relationship between mental health and cardiovascular disease showed increased cardiovascular risk in response to certain post-traumatic stress disorder symptom clusters[64,65]. Furthermore, post-traumatic growth (beneficial psychological effects following significant trauma[46]) supported better cardiovascular outcomes[66], offering potential for future interventions. ADVANCE found that the mental healthcare provided within military systems was felt to be inadequate by Injured participants undergoing

rehabilitation[43]. A high rotation of mental healthcare providers led to disrupted therapeutic engagement and a lack of clinical progress. More generally, participants maintained that visible injuries were seemingly prioritised by military healthcare professionals over invisible ones[43]. These lessons are important to learn, to ensure that mental health is addressed for all those sustaining conflict injuries to avoid secondary injuries.

Given the increased prevalence of knee osteoarthritis at Baseline and increased risk of incident knee osteoarthritis at Follow-up 1 in participants with lower limb loss, we wanted to understand underlying mechanisms using molecular biomarkers[27,40]. These biomarkers, measuring inflammation, metabolism or cartilage and collagen turnover, offer the potential to identify who might develop osteoarthritis after injury and who might be at increased risk of pain or reduction in function. Early work suggests that those with or at increased risk of pain could be identified[67] while providing

**Fig. 5 | ADVANCE cohort participant quotes.** Pseudonymised quotes from participants taking part in ADVANCE-INVEST, which investigates the experiences and outcomes of transitioning from military to civilian life.

**A**

*"I've got a friend of mine who lost both his legs and he says, 'It's the best thing that ever happened to me was losing my legs'... It's all the support; everything is laid on a plate for all of them but those with lesser injuries or those that don't want to speak up and say, 'I'm having a problem here' or whatever, they're just left as if it's just a normal, 'Well, right go and see your GP'."*

(Eddie, injured without limb loss)

**B**

*"Initially, I was diagnosed with just a fractured wrist even though I kept telling them that I had neck pain, back pain and headaches. The end result skipping forward a couple of years after I kept going to see the doctor to tell him there was something wrong with me and they told me there was nothing wrong with me I was diagnosed with chronic traumatic migraine, mild traumatic brain injury, fractured neck and a brachial plexus injury."*

(Harry, injured without limb loss)

**C**

*"They are really, really quick to get you on to 'stubbies', get you onto the microprocessor knees...and then just get out and about on the programme that they've got to learn to walk on stairs and slopes and rough terrain and all that sort of stuff... My whole process was really quick because I think it was only two years give or take from when I went from to rehab to when I got discharged."*

(Joel, injured with limb loss)

potential underpinning mechanisms[68]. However, their role in the accurate prediction of future osteoarthritis requires further examination[40]. A concurrent analysis of the participants' proteome might provide this insight, as well as providing data on the future cardiovascular risk and conflict-related impact on biological ageing.

Patient-reported outcome measures associated with musculoskeletal pain[27,37,40] were expanded to comprehensively consider the biological, psychological and social factors associated with acute, chronic, nociceptive, and neuropathic pain to build on early pain research and mental health[69]. Those with moderate-severe pain had higher levels of anxiety, post-traumatic stress disorder and depression, compared to those with no-mild pain[69]. We further examined how conflict injury affects post-service employment, in particular, the potential roles of pain and mobility. We found that, although there was no overall difference in employment levels between the Injured and Uninjured groups, employment was influenced by pain and mobility in the Injured without limb loss group.

Our osteoarthritis research proposed different knee health trajectories for different osteoarthritis causes[40]. We describe a steeper downward trajectory for those with lower limb loss. However, this evidence did not discriminate between unilateral/bilateral limb loss, level of limb loss, or differences between the residual or intact-side limbs. A sub-analysis of participants with unilateral transtibial lower limb loss showed increased knee osteoarthritis on both sides compared to the Uninjured group, but worse on the residual limb 11 years post-limb loss[49]. Further work on other limb loss levels and on the contribution of gait biomechanics will be investigated.

ADVANCE data demonstrated that low back pain severity and functional disability were similar in those with conflict injuries, including limb loss. Low back pain was only higher in those with a comorbid spinal injury[37]. In addition, we demonstrated the contribution of body composition, race, prior low back pain, depression, current use of opioids, phantom limb pain, and residual pain to low back pain-related disability in people with lower limb loss[37]. This work provided insights into potential interventions, and how holistic measures required to treat low back pain in people with limb loss may be different to able-bodied people.

In addition to the expansion and collaboration of existing themes, new themes have been introduced, such as long-term outcomes associated with traumatic brain injury[55,70]. As a result, longitudinal blood sampling, MRI brain scans and neuropsychological assessment have been added, as well as additional utilisation of existing blood sampling for proteomic biomarker analysis, as referred to above. This work demonstrated that traumatic brain injury was present in almost a fifth of the Injured group, which was itself associated with worse depression, anxiety and post-traumatic stress disorder outcomes, increased risk of chronic pain and possible signals of future Alzheimer's disease[55].

Within ADVANCE, there has been a focus on those with limb loss, and whilst this has resulted in multiple impactful findings, it has left a group of participants with heterogeneous injuries. Some participants with less visible injuries expressed frustration that they were not prioritised by medical professionals to the same degree as those injured with limb loss (Fig. 5A)[43], with some feeling they had not been effectively medically assessed, diagnosed or treated at the point they were injured and therefore that they did not receive the appropriate treatment for their injuries thereafter (Fig. 5B)[43]. This is an important lesson to learn regarding post-conflict care to avoid secondary injury.

Exploratory unsupervised machine learning algorithms were tasked with identifying specific health outcome patterns in the Injured without limb loss group using musculoskeletal data[56]. This identified poor musculoskeletal outcomes in some participants who sustained head injuries, which will be investigated using traditional hypothesis testing methods in the future.

## ADVANCE cohort responsibility

Since its creation, ADVANCE has existed to develop knowledge to drive improvements in clinical care for those exposed to conflict, both military and civilians. As our knowledge grows, it is important to maintain the methodological validity of the longitudinal approach, whilst simultaneously promoting veteran health and driving appropriate clinical care and interventions. As well as feedback from clinical examination, participants are given a detailed signposting booklet which includes up-to-date information on a range of services available to military personnel, both serving and

**Fig. 6 | ADVANCE cohort participant quotes.**
Pseudonymised quotes from two ADVANCE participants (and co-authors) who are part of a larger Participant Panel.

**A**

*"The initial rehab I got wasn't, wasn't great…I wasn't an amputee or anything, so I kind of cracked on and I think I suffered for a couple of years after with my fitness or different things. So, you know, to go back every couple of years to get scientists, doctors to have a really good, thorough look at you is really good."*

(Ethan, injured without limb loss)

**B**

*"You feel like you're still helping them, and that purpose is the main thing for me as it helps me maintain a link to my past that I don't want to keep too distant. Now, I'm a civilian working in the telecoms industry, working for a company and a life that is very far removed from the military. But it was an important part of my life and the ADVANCE study maintains that link, which I think is also really important."*

(Alan, uninjured participant)

**C**

*"For every explosion on operations, where there was a seriously injured person or somebody killed, there would have been a number of smaller injuries that were still life-changing for those involved. Those that experienced that might really struggle with the mental health aspect because they weren't the main focus of that major incident because others may have died and some may have become multiple amputees, for example. So that kind of unexpected survivors thing really got me thinking."*

(Alan, uninjured participant)

---

veterans (https://www.advancestudydmrc.org.uk/signposting/). The introduction of new interventions, clinical feedback, and signposting within a longitudinal cohort could introduce bias and confounders, as well as introduce interpretation challenges regarding causal inference[71]. However, this should not be at the cost of innovation and interventions that can improve clinical outcomes. For these reasons, interventions are piloted for acceptability within small ADVANCE sub-groups and are not widespread. However, regardless of interventions, it is also essential to consider the potential for a Hawthorne effect where participants change their behaviour because they know they're being observed and are receiving data about their health[72]. Over time, as the cohort develops, this might become more obvious.

The ADVANCE cohort is entirely male because too few women sustained conflict injuries required aeromedical evacuation to a UK hospital to satisfy statistical requirements. However, with the guidance of the Independent Scientific Advisory Groups, steps are being taken to address this.

## Participant reflections

Within the ADVANCE cohort, we work with a panel of representative participants to discuss and refine research proposals, results, personal reflections, and pathways to impact. Within this review, two ADVANCE participants (and co-authors) reflected on why they take part. As noted in our retention strategy, the 'health MOT' was highly valued (Fig. 6A). Military personnel have a unique culture, and participants see ADVANCE as a way to continue to look after their colleagues and friends, even if they have left the military (Fig. 6B).

The ADVANCE cohort links them to an important time in their life and gives them a sense of purpose. They expressed ideas to improve cohort benefits for participants, but also a concern that involvement with ADVANCE could trigger delayed mental health problems through being continuously reminded of potentially traumatic events.

The participants had time to consider the circumstances of injury, the spectrum of injuries seen in ADVANCE, and the hierarchy of injury that

may have started at the point of injury, not just during rehabilitation and public perspective later on (Fig. 6C).

The participants were enthusiastic about the potential for the military impact of ADVANCE cohort findings in the UK and worldwide, particularly given the current conflicts in Ukraine. However, they felt that even more could be done to proactively share and action findings from ADVANCE at a grass-roots level, particularly in areas such as mental health, traumatic brain injury, and cognitive function. Participants were also excited by the prospect of interventions being developed as a result of ADVANCE findings, especially for serving personnel and those involved in the ongoing current conflicts, but they understood the need to retain cohort integrity.

## Conclusion

The ADVANCE cohort is contributing to knowledge of long-term physical and psychosocial outcomes in personnel who served in the Afghanistan conflict, including those who sustained previously unsurvivable physical conflict injuries. We have shown the evolution and impact of the ADVANCE cohort from core objectives to incorporating new and cross-cutting themes, deterministic hypotheses, and intervention development. The work of ADVANCE is helping us understand the long-term health outcomes of conflict injury, which will inform care and rehabilitation of military and civilian casualties affected by current and future conflicts and contribute to the development of new interventions to improve outcomes.

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

## Acknowledgements

We wish to thank all the staff at both Headley Court and Stanford Hall who helped with the ADVANCE study. Your contribution has been invaluable. A full list of staff, past and present, can be found here: https://www.advancestudydmrc.org.uk/staff-acknowledgements/. The ADVANCE Study is funded through the ADVANCE Charity. Key contributors to the charity are the Headley Court Charity (principal funder), HM Treasury (LIBOR Grant), Help for Heroes, Nuffield Trust for the Forces of the Crown, Forces in Mind Trust, National Lottery Community Fund, Blesma—The Limbless Veterans, and the UK Ministry of Defence. The funders of the study had no role in study design, data collection, data analysis, data interpretation or writing of the manuscript.

## Author contributions

All authors have approved the submitted version of the manuscript and agree to be personally accountable for the accuracy and integrity of their contributions. O.O.S., F.W., A.N.B., A.M.J.B., H.K., C.B., N.F. and P.C. are responsible for conceptualisation. S.S., A.N.B., A.M.J.B., H.K., C.B., N.F. and P.C. are responsible for methodology. O.O.S., F.W., F.B., D.D., R.M., H.B., A.V., G.E., W.A. and S.D.S. are responsible for writing—original draft.

A.N.B., A.M.J.B., H.K., E.M., E.C., S.S., C.B., N.F. and P.C. are responsible for writing—editing and reviewing. O.O.S. and F.W. are responsible for visualisation. A.N.B., A.M.J.B., H.K., C.B., N.F. and P.C. are responsible for supervision. A.N.B., A.M.J.B., H.K., C.B., S.S., E.M., E.C., N.F. and P.C. are responsible for project administration. A.N.B., A.M.J.B., H.K., C.B., N.F. and P.C. are responsible for funding acquisition.

## Competing interests

N.F. is the recipient of grants from the UK Ministry of Defence and the Office for Veterans' Affairs, consultation fees and support for attending meetings from Gallipoli Medical Research, a member of the Academic Advisory Board for the Office of Veterans' Affairs, a specialist advisor on the release of patient data for research for NHS England, the Director of the Forces in Mind Trust research centre, the Director of the King's Centre for Military Health Research at King's College London, and a Trustee for Help for Heroes. All other authors declare no competing interests. There are no non-financial competing interests to declare.

## Additional information

**Peer review information** The manuscript was considered suitable for publication without further review at Communications. A peer review file is available.

