## [Transparent Peer Review file · Communications Medicine]

Understanding long-term physical and psychosocial outcomes from conflict to rehabilitation through the ADVANCE cohort

Corresponding Author: Dr Oliver O'Sullivan

The manuscript was considered suitable for publication without further review at Communications Medicine.

This file contains author rebuttals in order by version.

Reviewer comment	Author response	Author action
Dear Dr Salvatico, Many thanks for your thorough and insightful review of our paper. Please find below our responses to your main comments. Minor changes/additions to your in-text suggestions should be clear in the tracked changes. Kind regards, Dr Fraje Watson & Dr Oliver O’Sullivan		
Abstract		
Line 5: On health of individuals? How Healthcare systems are shaped by these changes? Both?	Thank you for this suggestion. Both of your suggestions are correct, but in the context of this paper, I have added just the health of individuals.	“Improvements in acute clinical care are often translated into regular practice, but long-term impacts on individuals’ health may not be appreciated until far later.”
Line 8: Related to soldier/ individual health outcomes	Thank you adding this clarity, I have incorporated it into the text.	“Most post-conflict data related to the health outcomes of military personnel are limited by retrospective methodologies examining isolated results or surrogate measures of health/well-being.”
Introduction		
Line 56: Is this relevant as the cohort was presented as military personnel not civilians - please clarify any role of civilian impact data on project.	Thank you for this comment. As you say, the ADVANCE study contains only military personnel. However, we feel that our work is increasingly relevant to civilians injured in contemporary conflicts because civilians are sustaining injuries via similar mechanisms during and post-conflict. Whilst the main impact of ADVANCE work will be military, we strongly believe there is value for civilian cohorts.	
Core objective outcomes		
Line 105: Please see Comment added below on page 14 about how section	Thank you for the suggestion. See our response below.	

could be modified slightly for better flow and organization.		
Line 112: Please introduce the “Injured” and “uninjured” groups more clearly before referencing in the results below. Perhaps with line 107 and 108 “From this point forward will be referred to as X in results”	Thank you for your comment. The Injured/Uninjured groups were defined in Figure 1 and in-text in the “Who are the ADVANCE cohort” section. However, additional clarity is always welcomed so I have added your suggestion in-text.	“The core objective of the ADVANCE cohort is to investigate the long-term physical and psychosocial outcomes of UK Service personnel and veterans who sustained serious physical conflict injuries during their deployment to Afghanistan compared to a frequency-matched comparison group (18); Injured and Uninjured groups as previously described.”
Musculoskeletal		
Line 126: Please clarify if these outcomes are derived from physical exam or questionnaire reports from individuals - if from physical exam- what was examined for this aspect of health?	Thank you for this suggestion, I have added the requested detail in-text.	“To evaluate musculoskeletal health, the physical examination included knee and hip radiography, full-body dual emission x-ray absorptiometry, and self-reported questionnaires (18).”
Line 128: Assume this is reported, not measured	Yes, correct, thank you for clarifying.	
Mental health		
Line 141: As measured by XYZ exam or self-reported by individuals	Thank you for this suggestion. I have clarified this in-text.	“Using self-reported questionnaires, we found an increased likelihood of generalised anxiety, depression, post-traumatic stress disorder, suicidal ideation, and mental health multimorbidity in the Injured group compared to the Uninjured group (41, 42).”
Outcomes for participants with limb loss		
Line 172: As measured/ reported as X	Thank you for your suggestion, this information has been included in-text.	“Social support, as measured by the Multidimensional Scale of Perceived

		Social Support, was similar in the Injured and Uninjured groups.”
Additional data sources		
Line 235: Is this a similar project- please briefly describe since results from this continue below. Was this dataset combined with ADVANCE for any research aim?	These are all raw military datasets collected for internal medical use and strategic use. For example, the JTTR data is collected by trauma nurses on-site as soon as a casualty arrives in the first military hospital (e.g., Camp Bastion in Helmand Province, Afghanistan). ADVANCE can access these datasets to find out information about the participant’s immediate trauma care, e.g. injury details, Glasgow coma scales score, litres of blood and blood product received, but they do not provide any outcome data. All outcome data is collected prospectively by ADVANCE. I have altered and added some wording to make this clearer.	“Additional data linkages enhance understanding of the impact of immediate and early medical and surgical interventions on long-term physical and mental health, with implications for clinical policy and practice. For example, analysis of the Ministry of Defence’s Joint Theatre Trauma Registry and Medical Emergency Response Team datasets revealed...”
Line 243: citation	Thank you for spotting this, it is now added in.	
Collaboration and future research in ADVANCE to ask “why?”, “how?”, and “so what?”		
Line 274: ?	Thank you for picking up on this, I am aware it is a lesser known and counter-intuitive process. However, it is already defined in-text in an earlier section as follows, “Furthermore, those with limb loss reported a large degree of post-traumatic growth, whereby serious	

	trauma is followed by beneficial psychological effects (45).”	
Line 311: While this section is highlighting deeper insight into results mentioned previously - it could be a bit repetitive with the results listed under the objectives/ aims as well as here. Please consider in the very early part of the piece just listing the specific assessment goals and measurements and perhaps consolidating all of the findings for those aims here by category.	Thank you for such in-depth consideration of the context and readability of this article. The core objectives at the start discuss distinct results for each theme based on the primary objectives and directly align with aims, methods, equipment, and analysis described in the protocol paper (ref 18: Bennett et al, BMI Open, 2020). The latter section describes findings that result from the study’s evolution through collaboration across themes, additional equipment/data, and new directional hypotheses to enhance and expand on the primary results. Whilst I can understand this may feel like repetition, we feel it is important to the way the ADVANCE study is continuously evolving and increasing its dimensionality over time. To more clearly guide the reader through this process I have added some in-text phrasing.	“The main discoveries from the core objectives across each of these themes thus far are described below and illustrated in Figure 3. Since the core objectives were decided, the ADVANCE study has evolved to collect additional data on new and existing themes, conducted cross-theme analyses, and new directional hypotheses to expand on findings from the core analysis. Findings from the core objectives will be described here, and results from further work will be discussed latterly.”